# Development of a Miniaturized, Automated, and Cost-Effective Device for Enzyme-Linked Immunosorbent Assay

**DOI:** 10.3390/s25175262

**Published:** 2025-08-24

**Authors:** Majid Aalizadeh, Shuo Yang, Suchithra Guntur, Vaishnavi Potluri, Girish Kulkarni, Xudong Fan

**Affiliations:** 1Department of Biomedical Engineering, University of Michigan, Ann Arbor, MI 48109, USA; maalizad@umich.edu (M.A.); yashuo@umich.edu (S.Y.); 2Department of Electrical Engineering and Computer Science, University of Michigan, Ann Arbor, MI 48109, USA; 3Center for Wireless Integrated MicroSensing and Systems (WIMS^2^), University of Michigan, Ann Arbor, MI 48109, USA; 4Max Harry Weil Institute for Critical Care Research and Innovation, University of Michigan, Ann Arbor, MI 48109, USA; 5Arborsense Inc., 674 S Wagner Rd., Ann Arbor, MI 48103, USA; sguntur@arborsenseinc.com (S.G.); vpotluri@arborsenseinc.com (V.P.); girishkulkarni@arborsenseinc.com (G.K.)

**Keywords:** biosensing, ELISA, microfluidics, point of care, automation

## Abstract

In this work, a miniaturized, automated, and cost-effective ELISA device is designed and implemented, without the utilization of conventional techniques such as pipetting or microfluidic valve technologies. The device has dimensions of 24 cm × 19 cm × 14 cm and weighs <3 kg. The total hardware cost of the device is estimated to be approximately $1200, which can be further reduced through optimization during scale-up production. Three-dimensional printed disposable parts, including the reagent reservoir disk and the microfluidic connector, have also been developed. IL-6 is used as a model system to demonstrate how the device provides an ELISA measurement. The cost per test is estimated to be <$10. The compactness, automated operation, along with the cost-effectiveness of this ELISA device, makes it suitable for point-of-care applications in resource-limited regions.

## 1. Introduction

Point-of-care (POC) biosensors are transforming the landscape of healthcare diagnostics by enabling rapid on-site detection of diseases and biomarkers [1,2]. Traditionally, diagnostic tests require complex and costly equipment and highly trained personnel within centralized laboratories, leading to delays in obtaining results. In contrast, POC biosensors allow for immediate diagnostic feedback at or near the site of patient care, providing clinicians with real-time data that can guide timely medical decisions and interventions. These biosensors are compact, user-friendly, and often do not require extensive training, making them ideal for use in emergency rooms, clinics, field hospitals, and even home settings. These benefits not only enhance patient outcomes but also reduce the burden on healthcare systems by minimizing the need for hospital visits and laboratory infrastructure. Their applications range from glucose monitoring for diabetes patients to detecting infectious diseases such as HIV, malaria, and, more recently, COVID-19.

Enzyme-linked immunosorbent assay (ELISA), one of the most widely used and well-established biosensing techniques, relies on antigen–antibody interactions to detect specific biomolecules [3,4]. The simplicity, robustness, and specificity of ELISA have made it a cornerstone in clinical diagnostics and laboratory research. Traditionally, immunoassays based on 96-well plates in combination with optical intensity-based detection in signal generation methods such as colorimetry [5,6,7], fluorescence-based [8,9,10], and chemiluminescence [11,12,13] are widely used for sensitive and quantitative analyte detection in a broad range of applications [14,15]. Long assay duration, complexity, high price, and the large dimensions of machines as well as the high cost per assay can make them only suitable for laboratory testing [16,17,18,19,20,21,22,23,24,25,26], raising the need for POC ELISA-based biosensors [27,28,29].

Table 1 shows a summary of a few widely used commercial ELISA machines, and their prices, dimensions, and weights. Most of the machines operate based on the combination of robotic (mechanical) movement with pipetting. Dynex Agility System, which has a price of ~$170,000 and dimensions of ~1 m^3^, is one example that operates based on pipetting [30,31]. Some other machines such as ELLA operate using microfluidic chips with channels and microfluidic valves [32,33]. As shown in the table, the price of the ELISA machines is at least tens of thousands of dollars, leading to an immense need for the development of a cost-effective ELISA machine.

In this work, we design and implement a point-of-care, cost-effective, and automated ELISA-based biosensor. The machine itself costs less than $1200, which can be further reduced by industrial and mechanical optimizations for mass production purposes. The 3D printed disposable parts are made based on Stereolithography (SLA) 3D printing [42] using clear resin. The 3D printed disposable parts cost only a few dollars. The reagents can be stored and shipped within the disposable reservoir assembly disk, eliminating the need for laboratory preparation and expert operation. The ELISA setup has dimensions of approximately 19 cm by 24 cm, and the highest part is 14 cm. Our device is used to detect various concentrations of the interleukin-6 (IL-6) analyte as a model system with an R^2^ value of 0.9937. This work can pave the way for highly cost effective, point-of-care, automated, and sensitive ELISA-based biosensors, which can be used for field-deployable applications.

## 2. Materials and Methods

### 2.1. Materials

The Human IL-6 DuoSet ELISA kit (DY206) and the 25X Wash Buffer Concentrate (WA126) are purchased from R&D Systems, Minneapolis, MN, USA. The Blocker BSA (10% in PBS, 37525), the SuperBlock Blocking Buffer (37517), the Pierce Streptavidin Poly-HRP (21140), the Poly-HRP Dilution Buffer (N500), and the SuperSignal ELISA Femto Substrate (37074) are purchased from ThermoFisher Scientific, Waltham, MA, USA. The Human Serum (H4522) is purchased from Sigma Aldrich, St. Louis, MO, USA. The Clear Resin V4.1 cartridge (RS-F2-GPCL-04) was purchased from Formlabs, Somerville, MA, USA. Step motors, linear actuator, Arduino UNO microcontrollers, step motor drivers, peristaltic pump, and all other electrical parts were purchased from Amazon.com, Inc., Seattle, WA, USA. The camera lens (FUHF125SA1) was purchased from B&H, New York, NY, USA. The camera (MU530-BI-CK) was purchased from AmScope, Irvine, CA, USA. Polystyrene capillaries were purchased from Optofluidic Bioassay LLC., Ann Arbor, MI, USA. See details of the mechanical and optical parts in Table 2.

### 2.2. Sandwich ELISA Protocol

The sandwich ELISA protocol used in this work is shown in Figure 1. It includes two parts, preparation and detection.

The preparation part involves the following steps. (1) Capture antibody incubation for 60 min, followed by washing. (2) 3% blocker buffer bovine serum albumin (BSA) in 1X phosphate-buffered saline (PBS) incubation for 30 min to block the unbound sites of the reaction chamber, followed by washing. (3) Superblock blocking buffer incubation for 15 min, followed by washing. The total preparation part takes about 105 min and is performed before the detection part using the same machine and disposables (disks, microfluidic connector) described previously.

The detection part involves the following steps. (1) Sample incubation for 20 min, followed by washing. (2) Detection antibody incubation for 20 min, followed by washing. (3) Horseradish Peroxidase (HRP) enzyme incubation for 4 min, followed by washing. (4) Substrate addition and incubation for a few seconds, followed by chemiluminescence signal detection. The total time for the detection part is ~45 min.

### 2.3. Design of Components

The overall design of the biosensor configuration is shown in Figure 2. It consists of reagent reservoirs on a disk, a microfluidic connector, connecting a selected individual reservoir to an ELISA reaction chamber, and the reaction chamber itself. The microfluidic connector can move up and down vertically to connect and disconnect the reservoir to the reaction chamber. Hence this microfluidic connector acts like a microfluidic valve, except that its operation is through its mechanical movement. Through the microfluidic connector, the sample/reagent fluids are pulled from the reagent reservoir using the pump. Upon arrival of the sample/reagent fluid into the reaction chamber, the pump stops, and the sample/reagent liquid stays within the reaction chamber for incubation. Once the incubation time is over, the pump continues pulling the sample/reagent to exit the reaction chamber into a waste. Finally, the signal is taken by reading the chemiluminescence intensity on the transparent reaction chamber, which is eventually translated into analyte concentration.

Figure 3 shows the detailed configuration of the reagent reservoirs and the microfluidic connector. The reagent reservoirs are made on a disk and each of them is designated for storing a specific reagent used in the ELISA protocol or used for adding a sample under test. Through the rotation of the reservoir disk, the desired reservoir is placed on top of the microfluidic connector. Figure 3b,c show the top and the side view of the biosensor setup, respectively. The vertical (up and down) movement of the microfluidic connector and the rotational movement of the reservoir disk are the only two mechanical movements for the operation of the biosensor, which can be implemented easily and cost-effectively using a linear actuator and a step motor, respectively.

Figure 4 shows the CAD design of the reservoir disk, as well as photographs of its 3D printed version. The disk has a diameter of 112 mm and contains 12 wells (top opening diameter = 4 mm and bottom exit diameter = 3 mm, see details in Appendix A) that can store reagents and receive samples. The notch underneath the reservoir disk is also shown, which is used for mounting the disk onto a step motor. Usually, each reservoir well holds a volume of <50 μL for reagents (a smaller volume can also be accommodated—see discussions in “Conclusion and future work”). The bottom opening of the reservoir well is sealed with aluminum foil before the reagents are added (note: in the current work the top opening is not sealed for simplicity, but in the future commercial version, it can be sealed. The top opening for the well designated to receive a sample is always left open).

The disk has a special design for wash buffer inlet to receive wash buffer from a flexible tubing placed right above it. The flexible tubing is connected to a large wash buffer reservoir (such as a 50 mL conical centrifuge tube). The outlet of the wash buffer is on the same circumference as other outlets for reagent reservoir wells. The inlet and outlet for wash buffer are never sealed. The reason to have a separate wash buffer reservoir and the associated inlet/outlet is to provide ample wash buffer for thorough rinsing, since there are often dead volumes in the microfluidics (such as microfluidic connector and ELISA reaction chamber) that have residual samples and reagents that affect ELISA measurements. The disk design and the buffer arrangement are highly flexible. The number of wells on the disk can be increased or decreased, depending on the needs. Additionally, if multiple types of buffers are needed, we can have multiple buffer reservoirs (i.e., several large tubes) and their corresponding inlets and outlets on the disk.

Figure 5 shows the CAD design of the microfluidic connector for 3D printing, along with a photograph of its 3D printed version. Since the bottom of the reservoir well is sealed, in order to release the reagent in the reservoir, the seal at the bottom of the reservoir can be pierced by the needle in the middle of the microfluidic connector, which can be 3D printed together with other parts in the microfluidic connector. Upon piercing through the seal of the reagent reservoir, the reagent fluid is pulled from the reagent reservoir using the pump through the holes on the side of the needle. In many other biosensing device designs, piercing a seal is accomplished with a permanent needle, which requires thorough rinsing between each step, thus involving many mechanical movements of the needles and dedicated mechanical components (such as a robotic arm to move the needle up and down to pierce into and pull out of the reagent reservoir, and to move the needle laterally for rinsing). In our design, no dedicated mechanical components are needed. The needle can be thoroughly rinsed by the wash buffer concomitantly with the rinsing of other microfluidic parts (such as microfluidic connector and ELISA reaction chamber, etc.).

### 2.4. 3D Printing

The CAD design of the 3D printable parts was created using the Autodesk Fusion 360 software (version 2.0.19941, Autodesk Inc., San Rafael, CA, USA), and the models were intended for printing on a Formlabs 3B+ 3D printer (Formlabs Inc., Somerville, MA, USA), which uses Stereolithography (SLA) technology. A Clear Resin V4.1 was used for both the reservoir disk and the microfluidic connector. The CAD designs were saved in the .stl format, and the print orientation along with other settings was selected using Preform software (version 3.33, Formlabs Inc., Somerville, MA, USA), developed by Formlabs, to modify the .stl file.

The overall design was optimized to avoid delicate features, which allows for the use of the lowest 3D printing resolution (0.1 mm) and minimizes printing time. For the 3D printing resolution of the microfluidic connector, the “adaptive” option was chosen. This has the second highest printing speed after the 0.1 mm resolution among all resolution options, and its printing time was similar to the one using the 0.1 mm resolution. The reason to choose the adaptive resolution was to ensure that the slightly delicate features such as the bending narrow channel within the microfluidic connector were printed out properly. After testing different resolutions, we observed that the 0.1 mm resolution could result in the possibility of narrow bending channels not being printed fully. Then the print file was uploaded to the printer using the Preform software.

The print platform area of the Formlabs 3B+ 3D printer has dimensions of 145 × 145 mm^2^. The fastest prints are performed when the structure’s wider features are aligned horizontally, in other words, when the number of printable layers in the vertical direction is minimized. This would require the reservoir disk to be placed laterally on the print platform in the Preform software, which results in not having enough space for another disk to be placed on the platform, since the diameter of the reservoir disk is 112 mm. Therefore, in the print platform of one printer, only one reservoir disk (with the current design) can be printed at a time. In this configuration, the print of each reservoir disk takes 3 h. However, printing multiple similar objects simultaneously can significantly reduce the printing time per unit. Therefore, in the future work, if a larger print platform (for example, UltiMaker 3D printer) is used, or if the dimensions of the reservoir disk is reduced (see the discussion in “Conclusion and future work”), a single platform can accommodate more reservoir disks, which would lead to a significant reduction in the per unit print time. Using the same platform of 3B+ Formlabs 3D printer that can fit 28 microfluidic connectors, it took approximately 280 min to complete the printing task.

A 3D printed disk is shown in Figure 4d,e. The thick disk was used to avoid potential bending during the UV curing as a part of post processing the 3D printed parts. Since the cost of Clear Resin V4.1 cartridge is $149.00 per liter, and the resin volume consumed for the print of each reservoir disk is 135 mL, the current cost is ~$20 per reservoir disk. Since each disk has the reagent reservoir wells that are enough to cover 3 sandwich ELISA runs. Therefore, the per test cost of the disposable disk is ~$6.7. In the case of competitive ELISA, each reservoir covers 6 tests, leading to a cost of $3.3 per test. A 3D printed microfluidic connector is shown in Figure 5d. The cost per piece is estimated to be $0.79 based on the total resin volume (148 mL at a price of $149 per liter) consumed to print 28 microfluidic connectors on the current platform.

Note that the cost of the 3B+ model of Formlabs 3D printer is ~$3500. Hundreds of thousands of pieces of disks and microfluidic connectors can be printed in the lifetime of the printer, and therefore, the contribution of the 3D printer to the cost of each disposable part is negligible.

### 2.5. Assembly and Operation of a Complete System

Figure 6 shows the schematic of the full map of the biosensor structure and how its components are assembled. As mentioned before, the rotational movement of the reservoir disk and the vertical movement of the microfluidic connector are the only mechanical movements required for the operation of our device. As shown in Figure 6, the reservoir disk is mounted on a step motor for its rotation, and the microfluidic connector is mounted on a linear actuator for its up/down movement.

Through a rotational movement of the step motor, the desired reagent reservoir moves to the top of the microfluidic connector. Then through the upward movement of the vertical linear actuator, the microfluidic connector moves upward until its needle pierces through the aluminum seal at the bottom of the reagent reservoir, which releases the reagent fluid into the microfluidic connector’s channel. Then the pump pulls the reagent into the system for a pre-determined duration and stops once the reagent arrives at the reaction chamber. Once the specific incubation time of the reagent is over, the microfluidic connector moves downward to be disconnected from the reservoir, and the reservoir disk rotates to the location of the wash buffer outlet.

The wash buffer outlet at the bottom of the disk is internally connected to an inlet at the top the disk (Figure 4d, inset), which is connected to a large wash buffer reservoir (a 50 mL conical centrifuge tube in our case) via a peristaltic pump, as shown in Figure 6. During rinsing, the peristaltic pump adds a few drops of the wash buffer into the microfluidic connector, until the tip of the needle is covered with the wash buffer (see Figure 5c), and after a few seconds, it is pulled by the pump through the reaction chamber to the waste. The above process can be repeated a few times until the desired amount of the wash buffer is used. The amount of wash buffer used in each of the wash steps was chosen to be a few hundred micro-liters to ensure a thorough wash of the reagents unwantedly accumulated in the dead volumes. In the end, after a few seconds of wash buffer incubation inside the reaction chamber, the wash buffer is fully pulled out of the system by the pump and the reservoir disk rotates to place the next reagent reservoir in the protocol on the top of the microfluidic connector.

The steps mentioned above are repeated until all the incubation steps are completed. Finally, the substrate reagent is pulled into the reaction chamber, where it generates a chemiluminescent signal, which is captured by the camera placed above the reaction chamber (shown in Figure 7c). During the course of the experiment, the disposable syringe attached to the pump collects all the waste fluids.

Although this study reports a prototype version of the device, future commercial iterations could include design optimizations such as O-ring reinforced connections to ensure long-term reliability. Even in its current form, however, the reservoir disk and microfluidic connector exhibited over 80% success rate in solution transfer, supporting the robustness of the approach.

Samples were introduced into the designated reservoirs through the top hole using a micropipette. Reagents were then delivered via a syringe pump operating at a constant flow rate of 40 μL/min, which effectively functioned as a flow injection system to ensure continuous and stable delivery. All assays were performed at room temperature. Comprehensive details of reagent types, concentrations, and volumes are provided in the Appendix A.

Figure 7 shows the completed and fabricated setup of the automated ELISA machine. Both the linear actuator and the step motor are driven by stepper motor drivers and are controlled using Arduino UNO microcontrollers. Figure 7a demonstrates the 3D view photograph of the complete setup. The dimensions of the area are 19 cm by 24 cm, and the highest part in the setup, which is the camera’s position, is about 14 cm. The electronic parts, including the microcontrollers, drivers, and the bread board for voltage distribution, are packaged on the corner of the setup shown in Figure 7a. The side view photograph is shown in Figure 7b. The small peristaltic pump used for pulling the wash buffer out of its reservoir and delivering it to the wash buffer inlet above the reservoir disk is also shown in Figure 7b. Figure 7c shows a close-up photograph of the reaction chamber placed underneath the camera.

### 2.6. Preparation of Disposables

In our current sandwich ELISA, reagents were prestored on each well, and the volume of the reagents can be <50 μL. The concentrations of each reagent used in our pre-coating and detection IL-6 ELISA protocol along with the estimated cost of their dilution are shown in Appendix A. As shown in the table, the estimated cost of reagents per test is $0.855.

Many types of microfluidic reaction chambers can be used on our system. For simplicity, in this work, we chose to use polystyrene capillary cartridges purchased from Optofluidic Bioassay LLC, which have been used previously in many ELISA applications [11,12,13]. Each capillary cartridge comes with 12 individual capillaries (~15 mm in length, 0.8 mm in inner diameter, ~2 mm in outer diameter, and ~8 μL in total chamber volume). We used only one of them for each test. The cost of each capillary is ~$1. Since the reaction chamber has only 8 μL inner volume, much less than the ~50 μL volume of the reagents used in the test, the entire chamber is filled with the reagents during the incubation. The capillary is connected to the outlet of the microfluidic connector upstream and to a pump downstream with flexible plastic tubings (see Figure 1).

### 2.7. Cost of Components

The total cost of the machine, including the linear stage, the step motor, peristaltic pump, microcontrollers, and motor drivers, is about $1200 (see Table 2), which can be reduced significantly by industrial level optimizations and mass production/purchase of the parts. It is noteworthy that the syringe pump used in our experiments is of commercial grade, which costs a few thousand dollars. However, using 3D printed clamps, and a linear actuator, we can build a homemade syringe pump, which would cost under $130 and can still be housed in the current enclosure.

It should be noted that while the bill of materials demonstrates the low component cost of the device, actual large-scale production would include additional expenses such as labor, overhead, and profit margins, which could increase the overall cost into the several-thousand-dollar range. It is also noteworthy that although a commercial syringe pump was used in this prototype, manufacturing a low-cost pump for broader distribution would require additional engineering to ensure reliable and repeatable performance.

## 3. Results and Discussion

### 3.1. IL-6 Detection

Although the developed ELISA biosensor is compatible with various assay formats, including direct, indirect, sandwich, and competitive ELISA, sandwich ELISA was selected in this study due to its greater complexity and number of procedural steps. This format was chosen to rigorously evaluate the system’s performance under demanding detection conditions.

The protocol used is described in the “Materials and methods” section. To evaluate the detection capability of the proposed biosensor, interleukin-6 (IL-6) samples with concentrations of 0, 100, 400, 700, and 1000 pg/mL were prepared and subjected to the same chemiluminescence imaging protocol. IL-6 plays a key role in the immune response, particularly in inflammation and infection [43,44,45].

Since chemiluminescence emission primarily occurs in the blue region of the spectrum, the captured images were split into red, green, and blue channels, and the blue channel was used for analysis (Figure 8a,b). A rectangular region covering all capillaries was drawn horizontally across the image using ImageJ (version 1.54, National Institutes of Health, Bethesda, MD, USA), and the gray value profile was extracted along the horizontal axis. ImageJ automatically averages the intensity vertically across the height of the rectangle, resulting in a one-dimensional profile plot with peaks corresponding to each capillary (Figure 8c). Each pair of peaks represents duplicate measurements for a specific IL-6 concentration, which are labeled above the corresponding capillaries in the image. For consistency, the center point of each capillary was selected as the measurement location, and these points are marked with red dots in the figure. The intensity values at these points were averaged for each concentration, and standard deviation error bars were calculated from the duplicates. A calibration curve was constructed using the resulting mean intensities (Figure 8d). The data were fitted using a four-parameter logistic (4PL) model, which yielded an R^2^ value of 0.9937. The fit follows the following equation:(1)$$y=313713.97+\frac{6.12-313713.97}{1+{\left(\frac{x}{3925459.28}\right)}^{0.89}}$$
where *x* represents the IL-6 concentration and y denotes the signal intensity count (0–255).

As visible in some of the capillaries in the figure, faint signals can occasionally appear outside the capillary region. This is likely due to light reflections caused by minor leakage of reagent into the outer reaction chamber during tubing-based fluid delivery. However, this had no effect on the intensity measurements within the capillaries, and accurate quantification was successfully achieved. To further assess applicability in real-world conditions, the biosensor was challenged with 100× diluted pooled human serum spiked with IL-6 at 400 and 700 pg/mL. Recovery rates were found to be 108.7% and 90.2%, respectively, confirming the system’s capability to operate effectively in complex biological matrices.

The limit of detection (LoD) was estimated based on the baseline signal from the 0 pg/mL sample and its standard deviation, using the formula LoD = baseline + 3σ. The corresponding signal threshold was mapped onto the calibration curve using the 4-parameter logistic (4PL) model, yielding an LoD of approximately 7.13 pg/mL.

### 3.2. Discsussion

For context, recent miniaturized IL-6 assays report comparable LoDs of 11.29 pg/mL [46] (optical LSPR microarray; ~40 min), 6 pg/mL [47] (modular microfluidic on-chip assay), 5 pg/mL [48] (colorimetric lateral-flow with silver amplification, serum), and 4.6 pg/mL [49] (label-free LSPRi).

Taken together, these results indicate that a mechanically simple architecture, based only on coordinated rotary and linear motions, combined with a compact set of 3D printed components, can achieve LoD values comparable to state-of-the-art IL-6 assays. The absence of multi-axis robotics and custom optics reduces the bill of materials, assembly time, and operational burden, which improves robustness and field deployability. Revisiting Table 1, our cost and part count are substantially lower than those of industrial counterpart systems, suggesting a favorable path to scale and translation without sacrificing performance.

## 4. Conclusions and Future Work

In this work, we present the details of developing a miniaturized automated, and cost-effective ELISA device with a small size and light weight that can be used for POC applications and in resource-limited regions. Detailed cost analysis for the device in Table 2 suggests that the actual cost can be as low as ~$500 if the expensive camera and lens ($803 altogether) are replaced with an inexpensive lens and a light detector. To further reduce the per test cost, a few approaches can be implemented in the future.

For reservoir disks, which account for the significant portion of the disposable cost, a highly durable 3D printing resin such as the Tough 2000 Resin ($149 per liter from Formlabs) can be used for more durable printing of much thinner disks for reduced material cost and higher printing speed. The reservoir disks can also be manufactured with injection molding, which significantly reduces the cost per piece.For microfluidic connectors, it would be difficult to use injection molding for mass production with the current design. We can stay with the current 3D printing method with more durable resin described above. Or we can re-design the microfluidics (along with the embedded needle) that are more amicable for injection molding.For reaction chambers, instead of using capillaries with a circular cross section, microfluidic channels (or chambers) fabricated on a planar surface can be implemented, which is more suitable for mass production. More reaction channels can be added to accommodate multiplexed detection.The dimensions of the reservoir wells and all microfluidic channels can be reduced further to reduce reagent consumption.

## Figures and Tables

**Figure 1 sensors-25-05262-f001:**
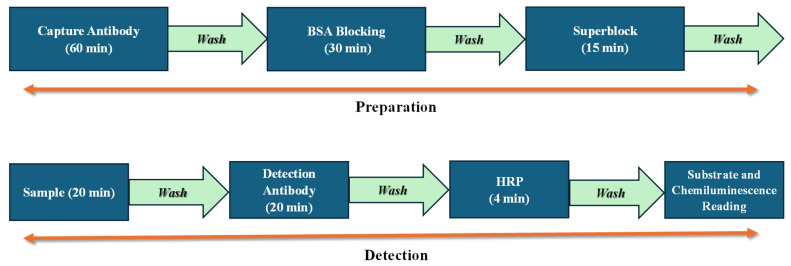
Schematic of the sandwich ELISA protocol used in our experiments.

**Figure 2 sensors-25-05262-f002:**
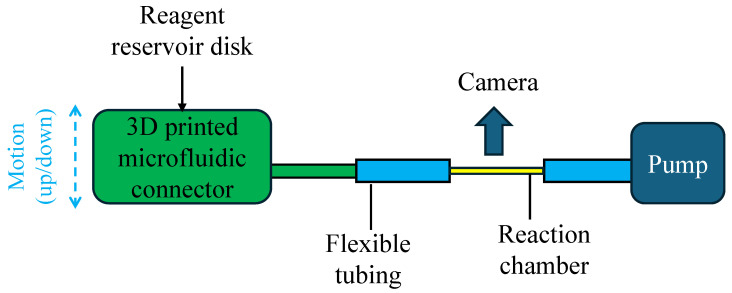
Overall design of the biosensor configuration.

**Figure 3 sensors-25-05262-f003:**
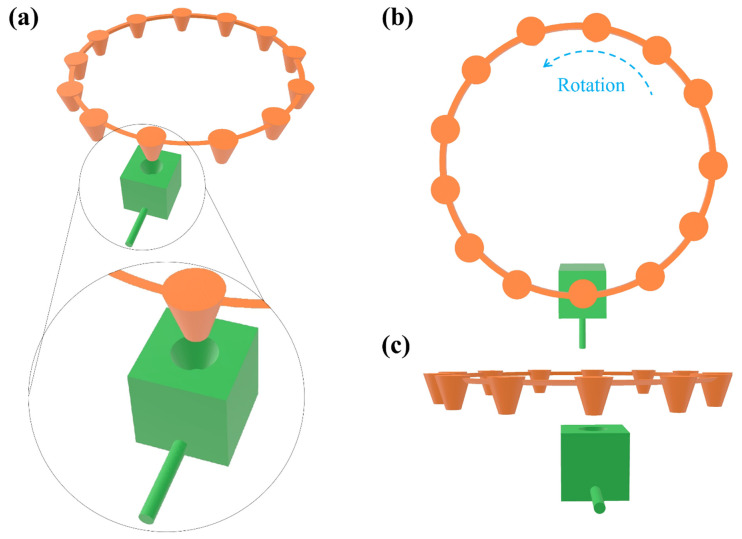
(**a**) Schematic of the biosensor configuration including the reservoir disk (orange) and the microfluidic connector (green). (**b**,**c**) top and side view of the biosensor configuration.

**Figure 4 sensors-25-05262-f004:**
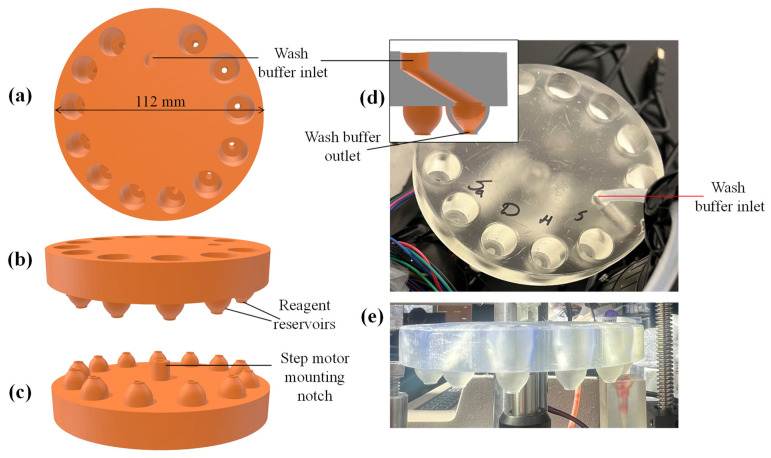
(**a**–**c**) Top, side, and bottom view of the CAD design of the reservoir disk for 3D printing. (**d**,**e**) Photographs of the corresponding 3D printed version. For a magnified view of a reservoir well, see Appendix A.

**Figure 5 sensors-25-05262-f005:**
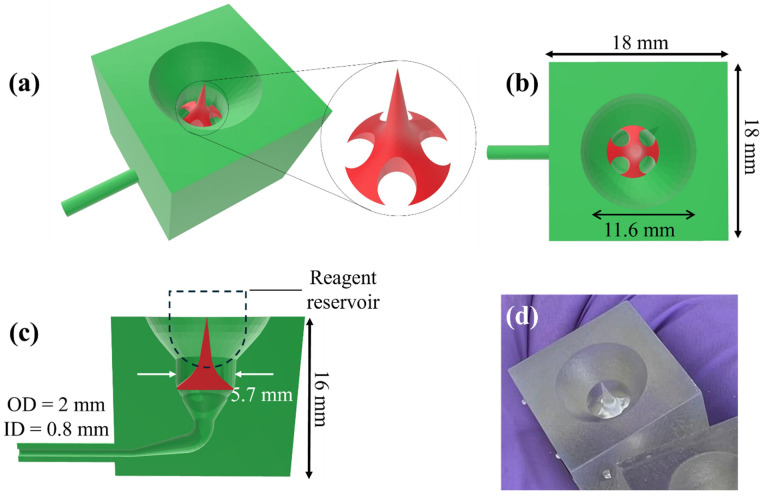
(**a**–**c**) CAD design of the microfluidic connector (green) along with the piercing needle (red) for 3D printing. The relative position between the needle and the reservoir well during seal piercing is shown in (**c**). (**d**) Photograph of its 3D printed version.

**Figure 6 sensors-25-05262-f006:**
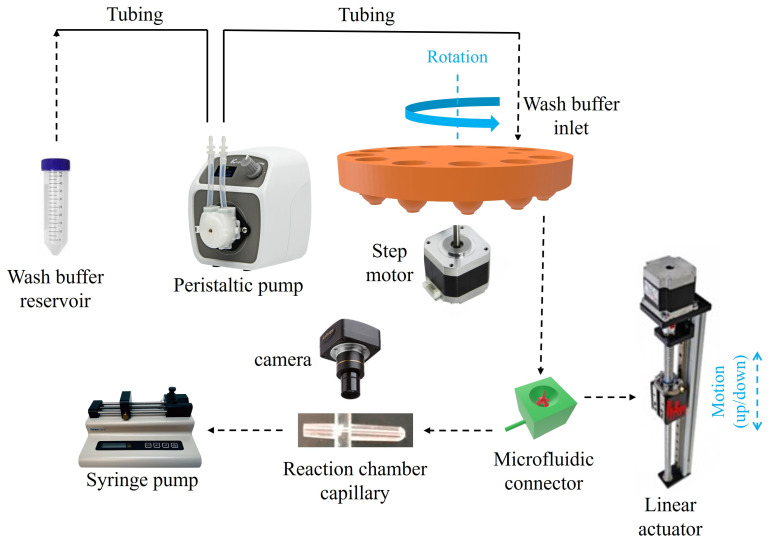
Complete map of the biosensor configuration with its parts. Note: Parts are not to scale; only relative positions and functional components are illustrated. Dimensions are provided in the text.

**Figure 7 sensors-25-05262-f007:**
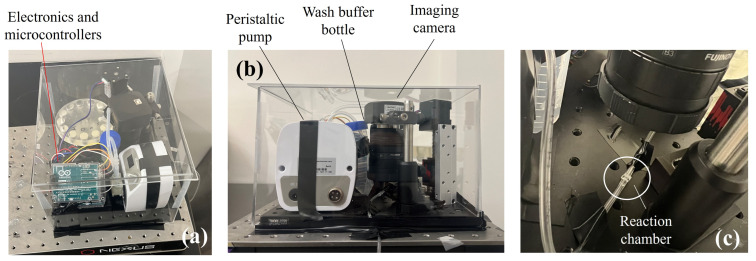
(**a**) 3D view of the complete biosensor setup. (**b**) Side view of the setup. (**c**) A capillary (reaction chamber) underneath the camera for chemiluminescence reading. The dimensions of the whole setup are: 24 cm × 19 cm × 14 cm (highest portion).

**Figure 8 sensors-25-05262-f008:**
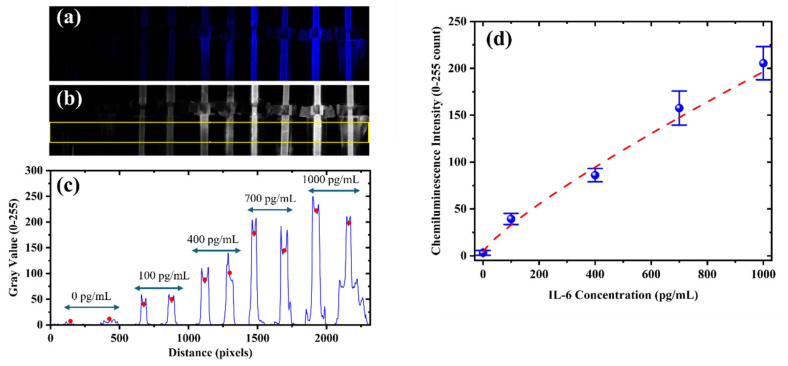
(**a**) Chemiluminescence images of IL-6 samples at 0, 100, 400, 700, and 1000 pg/mL, each tested in duplicate (left to right). (**b**) Blue channel extracted image with a yellow rectangle indicating the region used for intensity extraction. (**c**) Gray value profile (0–255) vs. horizontal distance (pixels), averaged vertically within the selected region. Each pair of peaks corresponds to duplicate capillaries, with red dots marking the center points used for signal quantification. Concentration labels are shown above. (**d**) Calibration curve showing mean intensity along with error bars representing duplicate measurements, fitted with a 4-parameter logistic (4PL) model (R^2^ = 0.9918, dashed line).

**Table 1 sensors-25-05262-t001:** Summary of the commercially available ELISA machines along with the machine reported in this work, comparing their price, size, and weight.

Biosensor Name	Price ($)	Company	Dimensions (cm)	Volume (L)	Weight (Kg)
ELLA [34]	~52,000	Bio-Techne	38 × 54 × 26	53	>20 (estimated)
DYNEX DS2 [35]	~73,000	DYNEX Technologies	54 × 68 × 66	242	48
DYNEX DSX [35]	~130,000	DYNEX Technologies	106 × 91 × 80	476	136
DYNEX Agility [35]	~170,000	DYNEX Technologies	90 × 123 × 125	1384	296
Crocodile ELISA [36]	~29,000	Berthold	~40 × 40 × 40	~64	>20 (estimated)
The Bolt [37]	~37,000	Axiom Medical Supplies	48 × 53 × 56	142	27
BIOBASE1000 [38]	~11,000	Biobase	93 × 69 × 86	552	130
Gyrolab xPlore [39]	172,000	Gyrolab	54 × 58 × 64	200	80
Gyrolab xPand [39]	385,000	Gyrolab	121 × 67 × 82	665	160
DAS APE IF ELITE [40]	~61,000	DAS	113 × 77 × 75	653	120
DAS AP22 ELITE [40,41]	~46,000	DAS	62 × 83 × 72	370	85
**This work**	**~1200** **(manufacturing cost)**	**-**	**19 × 24 × 14**	**<6.4**	**<3**

**Table 2 sensors-25-05262-t002:** Items used in the automated ELISA machine and their contribution to the total cost. The cost can be significantly reduced if the expensive camera and lens are replaced with an inexpensive lens and a light detector.

Part	Item Price ($)	Count	Total Price ($)
Arduino Uno REV3	27.60	2	55.20
TB6600 stepper motor driver	9.98	2	19.96
FUYU FSL30 linear stage	83.00	1	83.00
Iverntech Nema 17 stepper motor	12.99	1	12.99
Fujinon HF12.5SA-1 C-Mount 12.5 mm lens	268.00	1	268.00
AmScope MU Series 5.3MP CMOS C-mount microscope camera	534.99	1	534.99
Peristaltic Pump 12V dc Kamoer DKCP	66.00	1	66.00
Handmade syringe pump (estimated)	130.00	1	130.00
Wefomey 72W Universal Power Supply	18.99	1	18.99
Breadboard + wires (estimated)	<10.00	1	10.00
**Total**	-	-	**~1200**

## Data Availability

Data is contained within the article.

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
