# Peer review of "Development of a Miniaturized, Automated, and Cost-Effective Device for Enzyme-Linked Immunosorbent Assay"

_sensors, 2025, doi:10.3390/s25175262_

Round 1

Reviewer 1 Report

Comments and Suggestions for Authors

General Comments:

The present manuscript describes the development of a biosensor based on a Miniaturized Device for Enzyme-Linked Immunosorbent Assay (ELISA) aimed at medical applications and point-of-care (POC) diagnostics. The proposed system seeks to address the limitations of conventional ELISA by providing a simplified, cost-effective, and portable alternative. Interleukin-6 (IL-6) was selected as the model analyte. The figures and images included are generally appropriate and informative. However, the manuscript requires several improvements in terms of structure, methodological detail, and scientific discussion. Specific comments are as follows:

  • Manuscript structure: The current organization of the manuscript should be revised. The "Materials and Methods" section is too brief and lacks sufficient detail, while the "Results" section includes extensive methodological information that should be relocated to the appropriate section.
  • Materials and Methods: This section should be expanded to include key experimental details such as the volumes and concentrations of samples and reagents used, the mode of sample introduction (e.g., micropipette, syringe), the flow rate employed, and the operating temperature. Clarification is also needed regarding whether a flow injection system was utilized. Notably, some of this information currently appears in the Results section and should be moved accordingly.
  • Reorganization of content: Sections 3.1 to 3.3 (up to line 270), as well as the content from lines 288 to 303, would be more appropriately placed in the "Materials and Methods" section rather than in the "Results."
  • Figure 8: This figure is neither referenced nor discussed in the text. Please ensure it is properly introduced and analyzed within the manuscript.
  • Discussion and comparison: The discussion section should include a comparative analysis of the analytical performance parameters (e.g., LOD, sensitivity, response time) of the proposed method relative to other established or emerging IL-6 detection methods, particularly those employing miniaturized ELISA platforms.
  • References: While the cited literature is generally appropriate, the formatting of web-based references should be aligned with the journal’s citation style. Additionally, more recent references should be incorporated to enhance the relevance of the background and discussion.
  • Novelty and differentiation: The manuscript should clearly state how the proposed methodology differs from other existing miniaturized enzyme-linked immunoassays for IL-6 detection. Emphasizing the novelty and advantages of the proposed approach would strengthen the scientific contribution of the work.
  • In the section of Materials, the models of the step motors, linear actuator, Arduino UNO microcontrollers, step motor drivers, peristaltic pump, and all other electrical parts that were purchased from Amazon need to be added. Also, the city and country of the trademarks (f.i. R&D Systems) need to be added.

Reviewer 2 Report

Comments and Suggestions for Authors

The article presents a good automatization alternative for reading an ELISA test based on capillaries. The article clearly illustrates the costs of the parts which is indeed lower than the current alternatives available on the market. The article is generally well written and explained. Please consider the following before publication:

  • Could you please provide a literature review on what other such attempts have been made before.
  • The article successfully demonstrates the low component costs of the proposed ELISA reading device, but some of these costs appear to be underestimated. Production includes many other processes such as labour time, keeping the production line up and running, associated costs (paying for office space, paying for an accountant) and profit margins. I estimate that such a device can be produced in the range of 5000-10000 dollars. Maybe it would be worth mentioning in the article some of these considerations. Along the same line the syringe pump currently used, as the authors state, is a “proper” syringe pump. While clearly stated by the authors manufacturing a cheap syringe pump could pose problems on the reliable usage of the device. Could you please comment on this in the article.
  • The other major point would be important to comment on is some sort of failure rate for the reservoir disk and microfluidic connector. What is the fail rate here when transferring solutions. I guess given some numbers would be useful even if it would be we tried 1000 times and it never failed. This would further enhance the article.
  • Also, in figure 4 I would somehow include the capillary, the camera and the syringe pump. I think this would give a more complete picture of the device and actually, most important, would be to have an actual/real photo of your device with clear labels of different parts. This would improve the readers understanding of the device.

Round 2

Reviewer 1 Report

Comments and Suggestions for Authors

This work describes the development and validation of a biosensor based on an Enzyme-Linked Immunosorbent Assay (ELISA) test. The overall quality of the manuscript has improved, particularly in terms of the figures, the formatting of the references, and the inclusion of details regarding the working conditions. It is therefore worthy of publication.